# Beyond the BMI: Validity and Practicality of Postpartum Body Composition Assessment Methods during Lactation: A Scoping Review

**DOI:** 10.3390/nu14112197

**Published:** 2022-05-25

**Authors:** Caren Biddulph, Mark Holmes, Anna Kuballa, Roger J. Carter, Judith Maher

**Affiliations:** 1Centre for Bioinnovation, School of Health and Behavioural Sciences, University of the Sunshine Coast, Maroochydore 4558, Australia; mholmes@usc.edu.au (M.H.); akuballa@usc.edu.au (A.K.); jmaher@usc.edu.au (J.M.); 2Liaison Librarian, University of the Sunshine Coast, Maroochydore 4558, Australia; rcarter@usc.edu.au

**Keywords:** maternal nutrition, maternal body composition, anthropometry, lactation, breastfeeding

## Abstract

The assessment of body composition during lactation is an important indicator of maternal nutritional status, which is central to the overall health of the mother and child. The lactating woman’s nutritional status potentially impacts on breastmilk composition and the process of lactation itself. The purpose of this scoping review was to synthesize comparative studies that sought to validate various body composition assessment techniques for use in lactating women in the postpartum period. Using the PRISMA-ScR guidelines, a comprehensive, systematic literature search was conducted using Scopus, Web of Science, and PubMed. Eight comparative studies were included in the review, with data from 320 postpartum women. The design methodologies varied substantially across studies, and included a range of simple techniques to advanced multi-compartment models for assessing body composition. The validity and reliability of measurement tools must be considered alongside issues of safety, practicality, and appropriateness to guide the research design when applied to lactating women.

## 1. Introduction

The human reproductive period represents a nutritionally sensitive phase for the mother [1]. Maternal nutritional status during pregnancy and lactation is a critical factor in the overall health and wellbeing of both mother and child, and as such, it can be useful to assess body composition as an indicator thereof [2]. However, following the many physiological changes that occur in pregnancy [3], the postpartum body undergoes dynamic changes to return to a state similar to that of its pregravid condition. Such changes are reflected in maternal body composition, which transforms in response to various metabolic and hormonal signals [4]. Shifts occur in fat metabolism and distribution [5], and there are residual alterations in the hydration of tissues in pregnant and early-postpartum women [6]. Changes such as augmented visceral fat deposition show great interindividual variability [7,8]. These changes raise doubt as to the applicability and accuracy of applying common body composition techniques to postpartum women that have been validated in other population groups. Nevertheless, body composition assessment remains important in tracking postpartum nutritional status and designing health and lifestyle interventions targeted at this group of women. For example, a key focus area could be reducing the excess body weight and body fat resulting from inappropriate gestational weight gain, postpartum weight retention, and/or postpartum weight gain [7]. Addressing this intrapartum weight retention is important, as it has been identified as a modifiable risk factor for future obesity and related adverse health conditions [8,9]. This is especially evident in westernized countries, where the postpartum diet is often either unbalanced in terms of nutrient composition or contains macronutrients and/or total energy in excess of a mother’s requirements [1].

If a mother chooses to breastfeed, the need to understand her nutritional status is perhaps even more heightened. There is an interesting interplay between the individual changes in maternal body composition over the period of lactation and the breastfeeding pattern: exclusive breastfeeding seems to promote greater maternal weight loss over the immediate postnatal period, and thus may offer some protection against future weight gain and obesity [10]. The lactating woman’s nutritional status also affects the composition and metabolome of her breastmilk, and therefore the nutrition of the infant [11,12]. Studies indicate that the nutritional content of a mother’s breastmilk may be related to her postpartum body composition, perhaps even to a greater extent than to her dietary intake [13]. As it is well accepted that nutrition during the early postnatal period influences an infant’s risk for metabolic disease and obesity later in life [14], systems of monitoring and designing interventions aimed at promoting a healthy maternal body composition are essential.

Appropriate, valid, and reliable measures of maternal body composition are required to inform nutritional investigations, recommendations, and guidance. A selection of in vivo techniques and the models upon which they are based have been used in such investigations, ranging from indirect and simplistic measures to advanced volumetric assessments. Basic 2 compartment (2C) models divide the human body into fat mass (FM) and fat-free mass (FFM), and include approaches such as air displacement plethysmography (ADP), hydrodensitometry, and isotope dilution [15]. Some of these methods are suitable for use in field studies (for example, anthropometry and bioelectrical impedance analysis (BIA)), and others to a laboratory setting (such as ADP via the BODPOD machine) [15]. In three-compartment models, such as dual-energy X-ray absorptiometry (DEXA/DXA), the FFM mass is further separated into total body water and solids (protein and minerals) [16]. More advanced 4C models can be used to delineate body fat, minerals, protein, and water and are considered criterion methods [17]. More recently, advanced laboratory-based techniques, including computed tomography, magnetic resonance imaging, and quantification of trace elements, have been used [16,18]. Body composition assessment methods can be evaluated against a number of criteria, or “reference” methods, such as dual-energy X-ray absorptiometry and densitometry. These are based on multicompartment models and are used in lieu of cadaver analysis, which is considered the gold standard [16,19]. Despite the abundance of literature investigating maternal body composition with various outcomes, consensus on the most valid method of assessment to use in this population group is lacking. Beyond the unique physiological aspects, there are several practical considerations pertaining to breastfeeding women that will guide the most appropriate choice of techniques and models. This review aims to scope any comparative studies investigating the validity of body composition assessment techniques used in lactating women, and to comment on the most practical and appropriate methods available. This review is intended for health professionals, nutritionists, dietitians, and those working in the field of lactation research.

## 2. Materials and Methods

### 2.1. Study Design

This scoping review was planned and conducted using the Preferred Reporting Items for Systematic Reviews and Meta-Analyses guidelines extension for scoping reviews (PRIMSA-ScR) [20]. The review protocol was drafted using the “Preferred Reporting Items for Systematic Reviews and Meta-Analysis Protocols” (PRISMA-P) and has been made publicly available; reference: Biddulph, C., Carter, R., & Maher, J. (2022). Beyond the BMI: Validity and Practicality of Postpartum Body Composition Assessment during Lactation. A Scoping Review Protocol (University of the Sunshine Coast, Sippy Downs, QLD, Australia, 2022). The protocol is published on the USC Research Bank website (https://doi.org/10.25907/00129 (accessed on 20 April 2022)).

### 2.2. Identifying the Research Question

We aimed to address the following question: “Which are the most valid, accurate, reliable, and suitable body composition assessment techniques for use with postpartum, lactating women?”.

### 2.3. Search Strategy and Eligibility Criteria

Comprehensive systematic electronic literature searches were performed in March 2022 to identify relevant studies reporting on the assessment of maternal body composition in postpartum, lactating women. Recommended health and nutrition electronic databases were searched, including Scopus, Web of Science, and PubMed. The following search strategy was applied to terms listed within the titles, abstracts, and keywords of articles: ((lactating or lactation or breastfeeding) and (“body composition” or densitometry or “fat mass” or “body fat” or adiposity or “fat-free mass” or “total body water” or dexa or “dual x-ray absorptiometry” or dxa or bodpod or anthropometry or “skin fold *” or “lean body mass” or “fat distribution” or adp or “body density” or “bio electrical impedance *” or “bio-electrical impedance *” or “bioelectrical impedance *” or bia) and (women or mother * or woman) and (valid * or accurate * or reliab * or technique *)).

A research librarian (R.C.) advised on the search strategy, search terms, and eligibility criteria, and reviewed the searches according to the PRESS (Peer Review of Electronic Search Strategies) guidelines [21]. The review includes original research papers that reported on investigations into the validity of different body composition techniques used in lactating women. We selected studies in humans published as full-length peer-reviewed articles and excluded conference abstracts, editorials, letters to the editor, and case reports. Outcomes that were assessed included any comparative results of maternal postpartum body composition assessments in lactating women. There was no time exclusion on publication dates. All reviewing tasks were performed by two independent reviewers (C.B. and J.M.), with discrepancies resolved after consensus. The final search results were exported into Endnote^®^ reference management software, and duplicates were removed. The process was repeated for subsequent databases and sources, with articles sorted into folders and details captured as per the PRISMA flow chart (Figure 1). These results were supplemented by articles found using methods such as citation searching of relevant articles and reference list searching (’snowballing’). Formal methodological assessment did not take place, as we sought to map this body of literature as comprehensively as possible. However, the authors consulted the JBI Manual for Evidence Synthesis (Scoping Reviews) [22] for guidance on conducting the review.

## 3. Results

### 3.1. Synthesis

The search strategy identified a total of 367 records (Figure 1). Two additional articles were identified from the reference lists of the included publications. Based on title screening, 22 articles remained for abstract screening. Of the remaining articles, 15 met the stated inclusion criteria and were subjected to full-text assessment. In total, eight publications were included in the scoping review, with data from 320 postpartum women.

### 3.2. Analysis of Methodologies

#### 3.2.1. Study Design and Sample Characteristics

The papers reported on maternal body composition assessments using a variety of techniques and models. These included anthropometry (*n* = 5), hydrometry (*n* = 6), densitometry (*n* = 2), impedance analysis (*n* = 3), and dual-energy X-ray absorptiometry (*n* = 3), as indicated in Table 1. A summary evidence table of the included studies is provided as Table 2. The articles reflect somewhat infrequent attempts at validating body composition assessment methods in lactating women, being published between 1989 and 2016. Three studies were conducted North America [23,24,25], two in Africa [4,26], and three in Europe (all in Sweden) [27,28,29]. Most of the studies performed investigations on healthy postpartum women who delivered term infants, but one only included overweight/obese women [27] and another a cohort of HIV-positive women [26]. The sample size varied substantially between studies, ranging from *n* = 10 in the earliest study [25] to *n* = 70 lactating women [27]. The timing of assessments ranged from 2 weeks to 15 months postpartum, with three studies performing a series of longitudinal measures over the first year and a half [23,27,29]. Of the studies with only one cross-sectional measurement, two chose the early-postpartum period (first two weeks) [24,28] and the others at around 2–3 months postpartum [4,25,26]. Two studies assessed gestational body composition [24,28] and two also included pre-gestational measurements for comparison [28,29]. Most of the studies considered basic exposure variables such as maternal age, ethnicity, socioeconomic status, geographical location, health status, gestational age, parity, birth weight, and mode of delivery. A variety of ethnic backgrounds was represented in the selected studies, with Caucasian, African, and Hispanic women included. Only one study commented on observations made when comparing the body composition variables between lactating and non-lactating women, where those who were breastfeeding had relatively higher levels of fat-free mass (FFM) hydration and density [24].

#### 3.2.2. Anthropometric Assessment Methods

Basic external measurements of the body’s dimensions, including maternal height, body weight/mass, gestational weight gain, and BMI, were assessed by Ellegård et al. and Papathakis et al. [26,27]. BMI was classified using the standard equation of weight/height^2^ (Quetelet’s index; kg/m^2^) [30]. In the former study, overweight and obese participants were intentionally recruited as the aim was to validate impedance techniques in overweight/obese postpartum women [27]. In the latter, mean postpartum BMI values were also in the ‘overweight’ category, in both groups of positive and negative HIV-infection status [26]. The authors sought to report on the incidence of wasting in lactating HIV-positive women, but found little evidence of this according to the BMI classification. However, it was noted that overall, basic anthropometrics and BMI cannot give an indication of body fat percentage or distribution.

Skinfold measurements were performed in the majority of the studies included [4,23,25,26,29], though these were simply comparator measurements and more sophisticated techniques were also used. Measurements were performed at three to four sites (biceps, triceps, subscapular and, in some cases, midline suprailiac), and summed. Only one study included mid-upper-arm circumferences (MUAC) [26] and used the Durnin–Womersley predictive equations to describe maternal body composition [31]. They found that these equations underestimated FM and overestimated FFM in lactating women (*p* < 0.05 for both) when compared to the reference stable isotope method [27]. Also known as hydrometry, this method involves the dilution of the isotope, deuterium, and allows for the measurement of TBW and subsequent assumption of FFM [19].

#### 3.2.3. Two-Compartment Models and Techniques

Techniques that aim to separate the body into two components, namely fat-free mass (FFM) and fat mass (FM), may be more robust methods of assessment than indices such as the BMI. However, each method has its own limitations and is based on theoretical assumptions and certain principles, as discussed below.

##### Isotope Dilution (Hydrometry)

Advanced isotope dilution with deuterium oxide (2H_2_O) was used as the comparator method in all eight of the included studies [4,23,24,25,26,27,28,29], even in the earliest work by Wong et al. [25] in 1989. This reference technique is used to derive fat mass and fat-free mass from a measurement of total body water (TBW). However, this method incorporates an assumption of the hydration status of FFM, which, as mentioned, shows high variability over the reproductive period. A noticeable decrease in TBW in the postpartum period follows its increase during pregnancy [6,29]; nevertheless, this seems to settle by at least three months after birth [23] and can be used in validation studies. Isotope dilution is also used for assessing breastmilk volume output, and therefore potentially lends added value to lactation research [32].

##### Bioimpedance Analysis (BIA)

Based on the conduction of an alternating electrical current applied to the body, and assuming the body is a cylinder and is at normal hydration levels, this indirect method of assessment allows for the estimation of total body water (TBW) [33]. Multifrequency measurements (MF-BIA) provide additional information about actual water distribution in the body, and bioimpedance spectroscopy (BIS) represents a more advanced technique used to measure body composition. All three variations of BIA were used in the studies included in this review. Medoua et al. [4] used a standard hydration factor of 0.73 and tetrapolar electrode placement and found that the BIA-based predictive equations underestimated TBW and FFM, and overestimated body fat at the group level compared to the reference method (deuterium oxide dilution). Although they only assessed a relatively small cohort (*n* = 44) of lactating women in Africa, it is interesting that the authors noted higher biases when using Black-specific equations, and that FM was overestimated by all 12 predicative equations that were used [4]. Also in Africa, Papathakis et al. [26] found that BIS overestimated TBW by 5–6% compared to the same reference technique in 68 lactating women. The authors deemed this difference to be acceptable, and found good correlation between BIS and isotope dilution methods in terms of FM and FFM [26]. This agrees with findings from a study conducted in Sweden with 21 women at an earlier postpartum stage of 2 weeks. The authors assured that estimating the average ICW, ECW, and TBW in groups of women is acceptable when using the BIS method, although agreement with reference values may not be adequate at an individual level [28]. The most recent validation study in this review supports the impression that the bioimpedance devices used are too imprecise for individual evaluation. This implies that the method may be better suited for use in population-based epidemiological studies. In overweight/obese postpartum women, both BIS and, even more so, MF-BIA underestimated FM and overestimated FFM and TBW, although changes in TBW after a 3-month intervention were accurate, when FFM hydration seems to return to normal ranges [27]. Overall, validation studies with impedance methods indicate that average TBW estimates correlate well with isotope dilution methods in healthy normal-weight postpartum women [28], but lack validity in individual FFM measures in those who are overweight/obese and HIV+ [26,27].

##### Dual-Energy X-ray Absorptiometry (DEXA/DXA)

DEXA was evaluated in three papers [23,24,27] and is based on values obtained from the differential absorption of X-rays of two different energies. Butte et al. [23] performed a comprehensive comparison of various body composition assessment techniques at three postpartum timepoints (not all mothers lactated throughout). DEXA was performed using *QDR2000 Hologic* equipment to measure fat mass, bone mineral content (BMC), and total lean-tissue mass (LTM). FFM was calculated as LTM plus BMC. Overall, a two-compartment method as assessed by DEXA was found to be acceptable for use in postpartum women beyond the puerperium (around 6 weeks post-delivery). This conclusion was based on the observation that longitudinal changes in the FM and FFM estimates correlated well between methods and that alterations in maternal hydration, density, and potassium content of FFM were negligible by this time [23]. This guidance was reiterated by further work in which DEXA measurements taken at 2 weeks postpartum (15 ± 2 days), differed significantly from reference four-compartment methods, when FFM hydration and density had not returned to normal/nonpregnant values (0.73). It is interesting to note that, compared to nonlactating women, FFM hydration levels were higher, and FFM density was lower (*p* < 0.05 for both) in lactating women early in the postpartum period [24]. In a more recent validation study, TBW as measured by DEXA was deemed precise and accurate when compared with doubly-labelled water reference results at 10 weeks postpartum [27]. In fact, DEXA was used as a reference method in this study to validate impedance methods due to its ability to measure FM and FFM with good precision and limited concerns about radiation safety [27]. However, a special consideration in the use of DEXA is ethnicity: Black women have around 8% more bone mineral than Caucasian women, for example; thus, population groups must be factored into the assimilation of the results [24].

##### Densitometry

Underwater weighing (UWW), or hydrodensitometry, is considered a gold standard approach to determine total body density via the principle of water displacement [34]. However, UWW is based on a two-compartment model and is thought to be prone to error (typically around 2.5%) due to variations in the FFM hydration and density [33]. Longitudinal data from Butte et al. [23] revealed that body density (kg/L) is not constant in the postpartum period, and shows a significant progressive increase over a 3- to 12-month period (*p* < 0.001). However, when UWW was used in various models of body composition assessment, the overall longitudinal changes in FM and FFM over a 12-month postpartum period were similar, despite systematic differences between methods (including TBK, DEXA, and deuterium dilution) [23]. Of interest is the finding that the FFM hydration, density, and potassium content did not seem to differ significantly between lactating and nonlactating women at any time over the first postpartum year [23]. This is a significant finding, and further research comparing these two groups across various body composition assessment techniques are warranted. In a practical sense, it implies that nutritional investigations using DEXA could be applied broadly to all postpartum women, regardless of infant-feeding method. However, the change in mammary tissue size and composition [35], as well as the small amount of increased fluid content in the breastmilk of a lactating woman, may be factors to consider.

Air-displacement plethysmography (ADP) is an alternative method to UWW, but has not been used in the studies covered in this review. An example is the “BOD POD” (described in more detail by McCrory et al. [32]), which has been used in recent studies assessing the body composition of lactating women [11,33]. The technique is rapid and simple [10], but its accuracy has not been validated in lactating women at this time.

##### Other Techniques

Other less widely-used techniques include whole-body potassium scanning (TBK) and total body electrical conductivity (TOBEC). Both methods were used in work conducted in the USA and were reported to have low levels of precision [23,24]. TBK was converted to FFM using the factor 60 mmol/kg FFM (2.346 g/kg FFM) and therefore comprises a two-compartment model with FM. However, considerable error in the estimates of FM using this method was noted, particularly in the early-postpartum period, when compared to deuterium dilution, hydrodensitometry, and DEXA [24]. The TOBEC technique consisted of a series of rapid and non-invasive 30 s scans to estimate FFM (and therefore FM in a two-compartment model). In comparison with other methods used (including TBK, DEXA, and deuterium dilution), the results for FM from TOBEC were ranked the lowest, and the estimation of changes in FM between 3 and 6 months postpartum differed from those obtained by TBW measurements [23].

Only one study used magnetic resonance imaging (MRI) in the assessment of BC in lactating women, and reported body fat values that were significantly lower than those obtained from TBW by isotope dilution or skinfold measures [29]. Although MRI is able to assess regional fat distribution (in this case, changes in adipose tissue volume), and is validated for use in adults [36], it is expensive and requires specialized training and equipment based in a laboratory setting [6]. Further considerations for women of a reproductive age are the exposure to a magnetic field, and physical discomfort, such as claustrophobia or fitting in the scanner if body size is substantially increased post-pregnancy [6].

Other newer techniques such as quantitative magnetic resonance (QMR) with an EchoMRI system, three-dimensional photonic scanning (3DPS), and computed tomography exist for the assessment of body composition [36,37]. To our knowledge and at the time of this review, none have been validated for use in lactating women.

#### 3.2.4. Multicompartment Models of Body Composition

As mentioned, two-compartment (2C) models of body composition are based on assumptions such as the constant hydration and density of FFM [15]. This may lead to error in simple 2C measurements, particularly in dynamic situations such as over the female reproductive cycle when there is large variability in the water content of FFM [38]. In contrast, multicompartment models allow for the direct measurement of FFM mineralization, hydration, and body density, and should reduce errors due to inter- or intra-individual biological variation over time [23]. The three-compartment (3C) model allows for the assessment of body water and density, and incorporates FM, TBW, and fat-free dry tissue [15]. The four-compartment (4C) model incorporates FM, TBW, bone mineral mass, and residual protein, and is often used as reference method because it requires the addition of bone mineral density measurement [19]. Hopkinson et al. [24] found significant mean and individual errors (related, in part, to lactation status) when comparing 2C techniques to the reference 4C Fuller et al. model in 56 women (*n* = 38 lactating and *n* = 18 nonlactating) in the early-postpartum period. Butte et al. [23] argued that these variations in FFM hydration settled by 3 months postpartum, and thus concluded that 2C may be acceptable for use in women beyond the initial postpartum period. This comprehensive assessment included several models and methods, namely: TBW, UWW, SF, TBK, DXA/TOBEC (2C), Fuller 3, Siri 3 (3C), and Fuller 4 (4C). Systematic differences were observed among all models for FM estimates, but overall measures of the longitudinal change in FM did not differ significantly (*p* ≤ 0.05) [23]. The authors found the mean fractional FFM hydration to be comparable at 3, 6, and 12 months postpartum to the conventional coefficient of 0.73 [39]. Elsewhere in the literature, comparisons of the models indicate that due to its ability to control for the aforementioned variability in the hydration of tissues, the 3C model is more valid than a 2C model [40]. The 4C model may be more accurate due to the elimination of assumptions in the ratio of FFM to bone mineral [24]. However, in healthy men and women (non-lactating), the 4C model only had a small margin of additional advantage in terms of accuracy over the 3C model [40,41].

#### 3.2.5. Consideration of Confounders

In any nutrition assessment, consideration of the broader ecosystem in which the subjects exist, together with other factors that may have an association with body composition, should be considered. None of the included studies explored maternal dietary components or characteristics during lactation, other than one that was embedded in a lifestyle intervention study, but did not report on diet in the included paper [27]. Publications may note other participant characteristics such as socioeconomic status, particularly ethnicity and cultural practices, food security, overall maternal health (for example, HIV status [42]), and lactation performance or breastmilk volume output in lactation studies. This review focused solely on validation studies and, thus, did not seek to explore these factors. Research into the nutritional status of lactating women would do well to comment on known influencers of postpartum body composition, such as maternal age, gestational weight gain, parity, and the mental health status of the mothers [6,43].

## 4. Discussion

The objective of this review was to synthesize studies that have assessed the validity of various models and techniques used to measure maternal body composition in lactating women. We also identified some key issues of measurement, which may influence the accuracy and reliability of measurements of body composition in this group. We provide some commentary and recommendations on which methods would be the most useful and practical in the determination of the body composition of lactating women, and how this compares to perhaps more valid and accurate techniques that can be used in research settings.

The assessment of body composition in postpartum women raises a few complex physiological issues for consideration, such as changes in body tissue hydration [24], individual variability in residual body weight retention [44], and augmented visceral fat deposition [7,8]. Therefore, impedance methods may not be suitable in the early-postpartum period due to deviations in the composition and hydration of FFM, which differs further between lactating and non-lactating mothers. The predication equations for BC may not be valid in lactating women, as the raised hydration of FFM may mean that the electrical current dispersion in tissues are altered [4]. Hydration levels seem to return to normal ranges by 3 months postpartum (mean fractional hydration FFM = 0.73 and FFM density = 1.098 kg/L) [23,24]; thus, 2C models can be used in body composition studies in lieu of 3C and 4C models in both groups after this time. Although impedance methods are simple, inexpensive, and portable, they tend to underestimate FM and overestimate TBW (and therefore FFM) when compared with criterion methods such as DLW and DEXA in overweight/obese postpartum women [27]. However, if used longitudinally across a group, these methods may be appropriate [23].

Whole-body DEXA can be used safely in lactating women (with caution to exclude the infant from the scanning room due to exposure to a small radiation dose) and can provide overall as well as regional assessment of body composition. DEXA measurements are less dependent on the assumption of constant hydration and density of FFM, and tend to reflect more accurate and precise TBW estimates than the DLW method [23]. DEXA has been validated for the assessment of body fat, abdominal obesity, and lean body mass across a range of BMI/body fat percentages in adults [45,46,47]. Other, more advanced techniques such as hydrodensitometry, air displacement, and isotope dilution are more expensive, less convenient, and more suited to laboratory-based clinical research studies. Despite systematic differences between a range of 2C (BW, UWW, SF, TBK, DXA or TOBEC), 3C (TBW and UWW (Fuller 3, Siri 3)), and 4C (Fuller 4) based on TBW, UWW, and BMD (Fuller 4) models, no differences in detecting change in FM and FFM were noted when used in this cohort. Using multicompartment models did not reduce the range of individual variability in estimates of FM or FFM [23].

In terms of recommendations for lactating women in particular, there are added considerations for choosing assessment methods around practicality, validity, and safety (for example, when using radiation-based techniques) [48]. The authors suggest that privacy during testing and confidentiality around results should be ensured, the use of female assessors would be preferable, and considered feedback and follow-up with referrals for psychological and/or nutritional counselling should be standard when dealing with this often-sensitive population group. Inexpensive, simple, convenient, and non-invasive methods may be more appropriate for use in women over the reproductive cycle and lactation [28]. However, BMI and skinfold techniques may not be the preferred indicators of adiposity in the postpartum period due to lack of discrimination between body compartments and hydration shifts in FFM. Multicompartment models may be preferable for longitudinal FFM assessments, whereas DEXA could be indicated for FM measures. Regardless of the technique chosen, attention should be paid to using standardized protocols and following best practice guidelines [49]. Use of ratios such as FM/FFM and expression of height-normalized indices of body composition are of greater value than single variables, and should be used in reporting the results (for example, FFM index (FFMI) as FFM/length^2^ (kg/m^2^) and FM index (FMI) as FM/length^2^ (kg/m^2^)) [50,51].

Additional comparative studies are required for emerging and more advanced body composition technologies to be validated for use in lactating women. Study designs should include longitudinal measures across an extended postpartum period, diverse and multi-ethnic samples, and comparator groups of non-lactating women. Such comparisons require appropriate analysis beyond correlations, using approaches such as Bland–Altman to determine the agreement of estimates between the new technique and the reference method [52].

Future research should focus not only on improving the measurement of body composition changes, but also understanding the predictors or modifiers of these changes across the reproductive cycle, with the ultimate goal of improving maternal and offspring health. Comprehensive nutritional assessments that incorporate dietary intake analyses, mental and general health assessments, metabolic markers, and the woman’s ecological setting alongside body composition measurements are warranted.

### Strengths and Limitations

Selection bias may have been introduced because only original journal articles published in English were included, and therefore the review only examines a limited number of scientific papers. Despite these limitations, this review is strengthened by its compliance to the PRISMA-ScR guidelines and robust search strategy conducted in multiple databases.

## 5. Conclusions

The purpose of this review was to synthesize comparison studies that sought to validate body composition assessment techniques for use in lactating women in the postpartum period. This review provides both technical and practical commentary on the available techniques, providing guidance for researchers considering body composition measurement in postpartum lactating women, as well as for clinicians working with this population. The choice of method and model centers around factors such as simplicity, safety, accuracy, and acceptability to this cohort. The determination of maternal nutritional status is important, as it is central to the overall health of the mother and child, and potentially impacts on breastmilk composition and the process of lactation itself [13,53]. The measurement of body composition during lactation gives insight into a woman’s nutritional status, and will guide the design and monitoring of lifestyle interventions aimed at postpartum women. The timing of measurements relative to parturition, and the longitudinal repetition of measurements over time, are critical factors to consider during this life stage. These factors will influence both the reliability and validity of measurements at the level of the individual.

## Figures and Tables

**Figure 1 nutrients-14-02197-f001:**
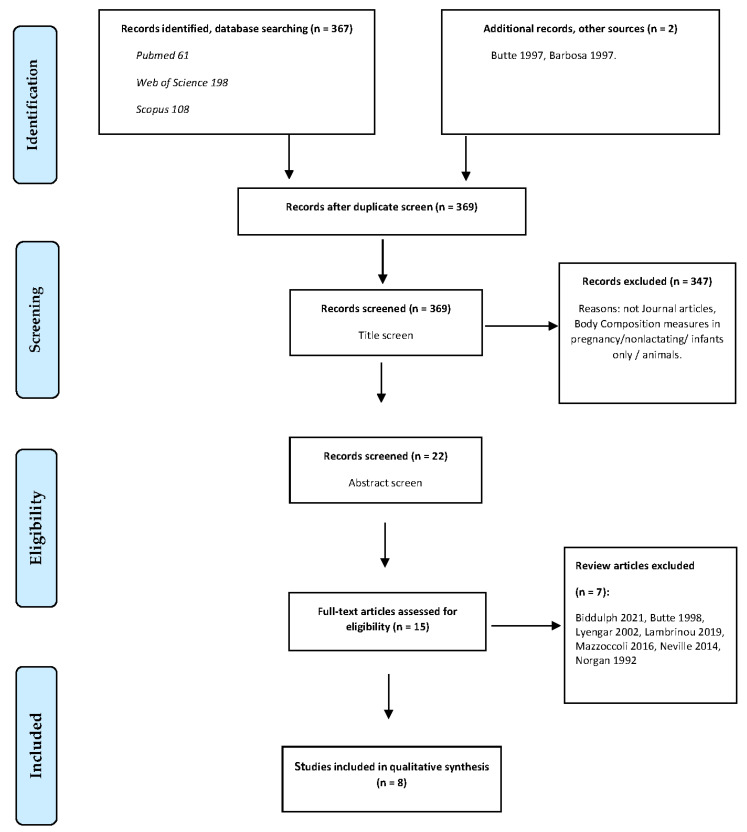
Preferred Reporting Items for Scoping Reviews and Meta-Analyses (PRISMA) flowchart describing study selection process.

**Table 1 nutrients-14-02197-t001:** Body composition assessment techniques used in the included studies.

Reference	Body Composition (BC) Assessment Methods (Maternal)
Butte, N.F., et al. (1997) [23]	Two-component (TBW, UWW, SF, TBK, DXA/TOBEC), three-component (TBW and UWW -Fuller 3, Siri 3), four-component models (TBW, UWW, and BMC-Fuller 4). TBW by deuterium oxide dilution.
Ellegård, L., et al. (2016) [27]	BW, height, BIS, MFBIA, DXA, and TBW via DLW.
Hopkinson, J.M., et al. (1997) [24]	TBW, TBK, body density, BMC by deuterium dilution, whole-body potassium counting, hydrodensitometry, and DXA (postpartum only).
Lof, M., et al. (2004) [28]	BIS (supine), reference isotope, and bromide dilution (only 2H_2_O was given at the postpartum measurement).
Medoua, G.N., et al. (2011) [4]	Anthropometry (triceps, biceps, subscapular, suprailiac sites), BIA (supine), reference method: deuterium oxide dilution.
Papathakis, P.C., et al. (2005) [26]	TBW using BIS and 2H_2_O to measure FFM and FM. Anthropometric measurements: BW, height, BMI, MUAC and four skinfold thicknesses (triceps, biceps, subscapular, and suprailiac).
Sohlström, A., et al. (1997) [29]	TBW by isotope dilution, MRI (30 transaxial images over the complete body except the head, hands, and feet), biceps, triceps, subscapular, and suprailiac SKF thicknesses.
Wong, W.W., et al. (1989) [25]	Anthropometry (triceps, biceps, and subscapular SKF) and deuterium dilution (HM, urine, saliva, breath).

BF, body fat; FFM, fat-free mass; FM, fat mass; BC, body composition; TBW, total body water; UWW, underwater weighing; SF/SKF, skinfold; TBK, total body potassium (whole body potassium scanning); DXA/DEXA, dual X-ray absorptiometry; TOBEC, total body electrical conductivity; BIA, bioimpedance analysis; BIS, bioimpedance spectroscopy; MF-BIA, multifrequency bioimpedance analysis; DLW, doubly-labelled water; MRI, magnetic resonance imaging.

**Table 2 nutrients-14-02197-t002:** Comparison studies assessing the validity of body composition assessment methods in lactating women.

Reference	Study Objective(s)	Study Population	Study Design	2C Body Composition (BC) Assessment Methods (Maternal)	3C Body Composition (BC) Assessment Methods (Maternal)	4C or Organ/Tissue Models for Body Composition (BC) Assessment Methods (Maternal)	Timing of BC Measurements	Method Comparison	Comparative Validity, Reliability, and/or Appropriateness	Relevant Findings	Strengths	Limitations
Butte, N.F., et al. (1997) [23]	To compare postpartum changes in BC by two-, three-, and four-compartment models and test for an effect of pregnancy or lactation on hydration, density, or potassium content of FFM.	N = 35 healthy postpartum women, lactating and nonlactating women (29 Caucasian, 2 African American, 4 Hispanic), USA.	FFM and FM estimated by nine BC models at three time points (3, 6 and 12 months postpartum) were compared using repeated measures ANOVA, as were changes in FFM and FM at two time intervals (3–6 months and 6–12 months postpartum).	TBW, UWW, SF, TBK, DXA/TOBEC.	TBW and UWW –Fuller 3, Siri 3.	TBW, UWW and BMC –Fuller 4; TBW by deuterium dilution.	3, 6 and 12 months postpartum.	Noted systematic differences among BC models in FFM and FM measures, but not in terms of longitudinal changes in FFM and FM.	No effect of pregnancy/lactation on the postpartum composition of FFM by 3 months, so two-component models are acceptable for use in postpartum women.	Pregnancy-induced changes in the hydration, density and potassium content of FFM were not evident by 3 months postpartum. Changes in FFM and FM did not differ significantly between models. The rank order from the highest to lowest estimate of FFM was TOBEC, TBW, Fuller 3, Siri 3, Fuller 4, UWW, SF, TBK, and DXA.	Numerous models and methods used: systematic differences noted between the models.	Not all lactating: breastfeeding prevalence was 27/35, 22/35, and 9/35 at 3, 6 and 12 months, respectively.
Ellegård, L., et al. (2016) [27]	To validate BIS and MFBIA with the reference methods DXA and DLW, and to assess BC in overweight/obese women postpartum.	*n* = 70 postpartum women, Caucasian, 35 overweight and 35 obese, Sweden.	Repeated measurements of BC (FM, FFM, skeletal muscle mass, and TBW) using reference methods and simple methodology (both cross-sectional and longitudinal).	BW, height, BIS, MFBIA, and TBW via DLW (isotope dilution of both D_2_O and H_2_O).	DXA	N/A	3 (baseline), 6 (12-week intervention) and 15 months postpartum (1-year follow-up).	Both BIS and, even more so, MFBIA underestimated FM and overestimated FFM and TBW.	Most accurate and precise TBW compared with DLW was obtained using 73% of FFM as assessed by DXA.	BIS underestimates FM, but accurately estimates muscle mass and changes in BC; MFBIA underestimates FM and overestimates TBW. Impedance devices underestimate FM in overweight and obese women.	Used accepted reference methods. Three repeated measures.	Systematic bias introduced via body positioning (impedance increases in supine over time). Bioimpedance devices used are too imprecise for individual evaluation.
Hopkinson, J.M., et al. (1997) [24]	To evaluate 2C and 3C models in the early-postpartum period.	*n* = 56 healthy women (38 lactating, 18 nonlactating), aged 19–35 years, USA.	Used a four-component model as a criterion for evaluating two- and three-component models.	TBW, hydrodensitometry.	DXA (postpartum only).	TBW, body density, BMC by deuterium dilution, whole-body potassium count (TBK).	Twice, at 36 ± 1 week gestation and 15 ± 2 days postpartum.	FM by TBK may differ by up to 6 kg from 4C model at 2 weeks postpartum. Use of standard 2C models to estimate FM results in significant error both at 2 weeks postpartum.	3C model compared favorably with 4C Fuller et al.’s model for estimation of mean and individual FM and change in FM.	At 2 wk. postpartum, FFM hydration and density had not returned to nonpregnant values, and differed between lactating (higher) and nonlactating women (*p* < 0.05). Standard TBW and body density estimates of FM differed from 4C estimates at both time points (latter was higher, *p* < 0.005).	Used a four-component model as a criterion for evaluating two- and three-component models.	Black women have around 7–8% more bone mineral and 4% more non-osseous mineral than white women, but this sample did not provide sufficient power to address the influence of ethnicity on intermethod differences.
Lof, M., et al. (2004) [28]	To evaluate BIS measurements of body water distribution in healthy women before, during, and after pregnancy.	*n* = 21 healthy women with healthy pregnancies and deliveries, lactating, Sweden.	Methodological study comparing BIS measures of ECW, ICW, and TBW, with reference methods, over the reproductive cycle.	BIS (supine), reference isotope and bromide dilution (only 2H_2_O was given at the postpartum measurement).	N/A	N/A	Before pregnancy, 14, and 32 weeks gestation, 2 weeks postpartum.	Average estimates of ICW by BIS were in good agreement with the corresponding reference data (not individual).	BIS (2C) may be useful for estimating average ICW, as well as changes in ICW, in groups of women during reproduction.	Postpartum average ICW, ECW, and TBW, estimated by BIS, were in agreement with reference data.	Longitudinal measures with reference methods.	Wrist–ankle measurements of resistance in BIS, assumes that the body consist of five cylinders.
Medoua, G.N., et al. (2011) [4]	To compare BC estimates using deuterium dilution, MF-BIA, and SKF techniques in lactating women.	*n* = 44 lactating women, aged 19 to 42 years, BMI 26.94 ± 3.61 kg/m^2^, Africa.	Comparison of the results of BC from the deuterium dilution technique with more convenient MFBIA and skinfold thickness methods.	Anthropometry (triceps, biceps, subscapular, suprailiac sites), BIA (supine), reference method: deuterium oxide dilution.	N/A	N/A	2.63 ± 1.31 months postpartum.	Inappropriate to use anthropometry or BIA equations to predict body composition in a population different from the population in which these equations were developed.	BC was affected (*p* < 0.05) by the technique used to measure it. Main factor implicated in the lack of agreement between BIA-predicted equations is lactation.	Anthropometric and BIA-based predictive equations overestimated BF by 2.7 to 11.7 kg; and underestimated TBW and FFM. Higher biases when using Black-specific equations.	Bland and Altman tests to determine bias and limits of agreement between values predicted by equations and measured by deuterium oxide dilution.	Small sample, no control group of healthy non-lactating women, use of the hydration factor 0.73.
Papathakis, P.C., et al. (2005) [26]	To determine the validity of BIS and anthropometric measurements to measure BC compared to the stable isotope dilution method. To describe the BC of HIV-infected lactating women.	*n* = 20 HIV-infected and 48 HIV-uninfected lactating women, 15–40 (median 24) years old, rural/low SES, African (Zulu), South Africa.	Compared the ability of BIS and anthropometry to determine TBW with isotope dilution (2H_2_O) to determine FM and FFM in HIV+ and HIV- breastfeeding women.	TBW using BIS and 2H_2_O to measure FFM and FM. Anthropometric measurements: BW, ht, BMI, MUAC, and four skinfold thicknesses (triceps, biceps, subscapular, and suprailiac).	N/A	N/A	Once, at 10 weeks postpartum.	Measurements determined by BIS correlated with 2H_2_O. BMI, MUAC, and skinfold-thickness measurements correlated strongly with FM measured by 2H_2_O; FFM only in HIV- mothers.	BIS comparable to reference 2H_2_O method. BMI and MUAC are useful in predicting FM, but are not valid measures of FFM in HIV+ mothers.	TBW by BIS was 5–6% greater than 2H_2_O method but FM or FFM did not differ significantly by method. Difference deemed acceptable by the authors.	Used the stable isotope deuterium oxide (2H_2_O) is a reference technique for measuring TBW.	Only assessed HIV-infected women without severe immune suppression.
Sohlström, A., et al. (1997) [29]	To compare changes in total BF assessed by MRI, body water dilution, and skinfold thickness in postpartum women.	*n* = 16 healthy postpartum women, lactating, Sweden.	Changes in total BF during the human reproductive cycle as assessed by MRI, body water dilution, and SKF thickness: a comparison of methods.	TBW by isotope dilution; biceps, triceps, subscapular, and suprailiac SKF thicknesses.	N/A	MRI (30 transaxial images over the complete body except the head, hands, and feet).	Before pregnancy and at 5–10 days and 2, 6, and 12 months postpartum.	Estimates of changes in BF by isotope dilution may be unreliable and invalid, as the degree of hydration of FFM may change over the course of the postpartum period in women.	Risk for bias when changes in TBF during reproduction are estimated by SKF-thickness technique and isotope dilution.	Changes in the degree of hydration of FFM may occur after delivery. SKF method tended to overestimate fat retention compared with MRI, and underestimate the amount of mobilized fat.	Results of BF from the MRI technique are more valid than changes estimated with use of body water or SKF techniques.	SKF technique: during both pregnancy and lactation, a redistribution of subcutaneous adipose tissue may occur.
Wong, W.W., et al. (1989) [25]	To compare estimations of BF, FFM, and TBW of lactating women by anthropometric and deuterium dilution methods.	*n* = 10 lactating women, aged 28.4 ± 4.2 years, USA.	Comparison of anthropometric equations and deuterium dilution method in calculating postpartum BC.	Anthropometry (triceps, biceps and subscapular SKF) and deuterium dilution (HM, urine, saliva, breath).	N/A	N/A	Once, at 3.4 ± 1.3 months postpartum.	Deuterium dilution method involves certain assumptions and errors, but is more direct and precise.	Difficulty in obtaining accurate and reproducible skinfold thickness measurements in the suprailiac regions of post-partum women, so excluded.	No significant differences in mean BF, FFM, or TBW between anthropometry and deuterium dilution; however, wide 95% CI, so not applicable to individuals.	Technical aspects of the deuterium dilution method also investigated.	Deuterium dilution overestimates TBW to the degree that deuterium exchanges with non-aqueous hydrogens.

BF, body fat; FFM, fat-free mass; FM, fat mass; BC, body composition; TBW, total body water; UWW, underwater weighing; SF/SKF, skinfold; TBK, total body potassium (whole-body potassium scanning); DXA/DEXA, dual X-ray absorptiometry; TOBEC, total body electrical conductivity; BIA, bioimpedance analysis; BIS, bioimpedance spectroscopy; MF-BIA, multifrequency bioimpedance analysis; DLW, doubly-labelled water; MRI, magnetic resonance imaging.

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
