# Peer review of "Beyond the BMI: Validity and Practicality of Postpartum Body Composition Assessment Methods during Lactation: A Scoping Review"

_nutrients, 2022, doi:10.3390/nu14112197_

Round 1
Reviewer 1 Report
Overview: This article, focused on body composition assessment during a highly critical stage in life that is associated with substantial changes and subsequent risk for later obesity and other comorbidities. The question of applicability of common body composition techniques during the postpartum period and lactation warrants further investigation. Overall, this scoping review was well developed and followed established publication standards for this type of publication. Presentation of key concepts, and summary of results would benefit from some refinement as follows:
Introduction: Although you mention the limitations of current approaches in body composition assessments, the description and overview of these come late, potentially leaving a reader without previous background knowledge uncertain of the main investigational outcomes as results are presented. Instead, a reorganization to include a section in the introduction, where common methods for body composition assessment are overviewed would be helpful from this aspect.
Tables/Figures: Table 1: Consider an alternate format for presenting areas of overlap (i.e. in technique) across articles. Additionally, consent formatting for all tables.
Conclusion: This section is limited in terms of pulling together major concepts from the data collection and reads more like a reiteration of the introduction. What were your major findings and how does this information inform suggested steps for further research and/or practice?
References: ~29/52 >50% of the articles cited were >10 years from publication. Although seminal work is appreciated, refinement of the supporting literature would strengthen this submission.
Author Response
Thank you for your time and valuable comments and suggestions. In response to your points:
Introduction: Expanded as suggested to equip the reader with some background knowledge regarding methods for body composition assessment.
Tables/Figures:
An alternate format has been included after consultation between all authors.
Conclusion: Thank you for this insight, this section has also been expanded to include pertinent yet succinct recommendations for practice.
References: We have reviewed all references again. Thank you for this comment; however, research reflecting actual validation studies is somewhat sporadic - perhaps indicating that it is a challenging area of research given the lifestage implications and that more research support is needed for studies in the space.
In terms of the studies included in the scoping review, criteria is set prior to the searches being conducted, as per the PRISMA guidelines referenced in the methods.
Finally, technical papers or those reporting on technical methods often still hold strong/are applicable once a method has been established. A clear example is reference 53- this statistical technique is still used in current publications globally.
Bland, J.M.; Altman, D.G. Statistical methods for assessing agreement between two methods of clinical measurement. Lancet 1986, 1, 307-310.
Reviewer 2 Report
Beyond the BMI: Validity and Practicality of Postpartum Body Composition Assessment methods during lactation. A Scoping Review
This study is a review, intended for healthcare professionals and those who are involved in lactation research. It attempts to identify the most valid, accurate and reliable body composition assessment technique(s) for lactating women in the postpartum period. This is important because nutritional status may dictate breast milk composition and may impact the future health of both mother and child. This study was conducted by electronic literature search in March 2022, using Scopus, Web of Science, and PubMed. In the end 8 peer-reviewed, published, full length articles were used, with data from 320 postpartum women. The methods for obtaining these data seem appropriate, although it would have been nice to have a larger sample size. It also seems that the available studies were quite varied, in terms of inclusion criteria, the time at which the researchers made their observations, and the observations they were making, which I would think would make it hard to draw conclusions based on available evidence. That said, the authors did a wonderful job of synthesizing the data and making recommendations on the various techniques that have been used to research body composition in this population. I especially appreciated the suggestion that privacy is important and that convenient and non-invasive methods be combined with counseling and support, as this is incredibly clinically relevant advice. I also appreciate the suggestion that future research focus on understanding predictors and modifiers, as well as improving measurement of body composition. Overall, I think this was a well-done review of the available literature and that it answers the research question.
Author Response
Thank you for your time and valuable comments and suggestions. We hope this will indeed be useful to others researching in this area, as it has been for our work. In response to your comments regarding sample size, we too would like to see greater numbers of validation studies and research support for nutritional investigations pertaining to postpartum women.